## Overview Review

climate change; biophysical systems; coastal ecosystems; sustainability

**Corresponding author:**
John W. Day;
Email: johnday@lsu.edu

# The coming perfect storm: Diminishing sustainability of coastal human–natural systems in the Anthropocene

John W. Day[1] 🄳, Charles A. Hall[2], Kent Klitgaard[3], Joel D. Gunn[4], Jae-Young Ko[5] and Joseph R. Burger[5]

[1]Department of Oceanography and Coastal Sciences, Louisiana State University, Baton Rouge, LA, USA; [2]College of Environmental Science and Forestry, State University of New York, Syracuse, NY, USA; [3]Department of Economics, Wells College, Aurora, NY, USA; [4]Department of Anthropology, University of North Carolina-Greensboro, Greensboro, NC, USA; [5]Department of Public Policy and Administration, Jackson State University, Jackson, MS, USA and Department of Biology, University of Kentucky, Lexington, KY, USA

## Abstract

We review impacts of climate change, energy scarcity, and economic frameworks on sustainability of natural and human systems in coastal zones, areas of high biodiversity, productivity, population density, and economic activity. More than 50% of the global population lives within 200 km of a coast, mostly in tropical developing countries. These systems developed during stable Holocene conditions. Changes in global forcings are threatening sustainability of coastal ecosystems and populations. During the Holocene, the earth warmed and became wetter and more productive. Climate changes are impacting coastal systems via sea level rise, stronger tropical cyclones, changes in basin inputs, and extreme weather events. These impacts are passing tipping points as the fossil fuel-powered industrial-technological-agricultural revolution has overwhelmed the source–sink functions of the biosphere and degraded natural systems. The current status of industrialized society is primarily the result of fossil fuel (FF) use. FFs provided more than 80% of global primary energy and are projected to decline to 50% by mid-century. This has profound implications for societal energy requirements, including the transition to a renewable economy. The development of the industrial economy allowed coastal social systems to become spatially separated from their dominant energy and food sources. This will become more difficult to maintain with the fading of cheap energy. It seems inevitable that past growth in energy use, resource consumption, and economic growth cannot be sustained, and coastal areas are in the forefront of these challenges. Rapid planning and cooperation are necessary to minimize impacts of the changes associated with the coming transition. There is an urgent need for a new economic framework to guide society through the transition as mainstream neoclassical economics is not based on natural sciences and does not adequately consider either the importance of energy or the work of nature.

## Impact statement

Although coastal zones comprise less than 5% of the earth's surface, they account for ~50% of the total human population and a large part of the economy. Coastal ecosystems are among the most productive globally, with high values of ecosystem services that exceed $10 trillion annually. The human population is concentrated in tropical coastal areas, which are strongly threatened by 21st-century megatrends. They are at the forefront of climate impacts, which include extreme warming, accelerated sea level rise, increasing frequency of extreme precipitation events and tropical cyclones, and changes in river discharge. Fossil fuels provide more than 80% of the world's primary energy currently and decrease to 50% by 2050. Transition to a renewable economy will require renewables to grow much more rapidly than fossil fuels did over the 20th century. The development of the modern economic system was underwritten by fossil fuels and the great source–sink functions of the biosphere. However, human activity has depleted the former and overwhelmed the latter. This series of overlapping issues are likely to coalesce in a perfect storm of impacts on humanity as it concentrates in an increasingly impacted and fragile coastal zone. Ironically, more energy will be required to address impacts of climate change and to compensate for and rebuild human and natural infrastructure located there. Coastal systems are provisioned by an energy-intensive global economy where production is often located far from where it is consumed. Transitioning away from fossil fuels, necessitated by depletion and to reduce climate impacts, will itself be energy intensive and will limit other sectors of the economy. All of this is likely to be felt most powerfully in the coastal zone, and society is largely unprepared for dealing with it. There is an urgent need for a new framework to guide society through the coming transition.

## Introduction

The world's coasts are generally areas of high biodiversity, biological productivity, population growth, and economic activity. Since about 2010, the global population has been more than 50% urban (Burger et al., 2019), and 40% and 50% of the world's population live within 100 and 200 km, respectively, of an oceanic coast. By comparison, the land area within 100 km of a coast makes up only 9% of the land area of the earth. Some regions are practically all coastal, such as the Malay Archipelago, the West Indies, the Pacific Islands, and most large island countries such as Japan, New Zealand, Madagascar, and the British Isles. Many of the world's great fisheries are associated with coasts, especially deltas (Pauly and Yanez-Arancibia, 2013). Deltas comprise a high proportion of the area of coastal ecosystems and about 400 million people live in or adjacent to deltas (Syvitski et al., 2022). A majority of the world's mega-deltas are located in tropical and subtropical coastal zones and most megacities are coastal, including Tokyo, Yangon, Kolkata, Dhaka, Ho Chi Minh City, Shanghai, Cairo, Karachi, Lagos, New York, Los Angeles, Rio de Janeiro, Sao Paulo, and Buenos Aires (Figure 1, Table 1). The globalized trade system is facilitated by large ports such as Shanghai, Rotterdam, Los Angeles-Long Beach, and the lower Mississippi River (Day and Hall, 2016). Most growth of population and economic activity in coming decades is projected to occur in coastal areas, especially in the wider tropics (Day et al., 2021; Kummu et al., 2022).

We develop in this paper how a suite of rapidly changing global forcings are coalescing in a "perfect storm" of cascading and interacting events and processes that will impact human society globally but especially humans now living in coastal regions.

Natural and human systems in coastal areas developed during relatively stable environmental conditions of the Holocene. This was especially the case for deltas that require stable sea levels and predictable freshwater and sediment inputs from drainage basins (Giosan et al., 2014; Day et al., 2019). However, shifting environmental and socioeconomic baselines such as climate change and energy security are threatening both human and natural systems (e.g., Colten and Day, 2018; Day et al., 2021). Changes in a variety of global forcings will severely threaten the sustainability of coastal ecosystems and populations. Central to these forcings is climate change. Over recent decades, climate change impacts have grown dramatically and are passing tipping points, many of which are likely irreversible (Zhang et al., 2020; Armstrong McKay et al., 2022; Jackson and Jensen, 2022; Kemp et al., 2022; McGuire, 2022). The global energy system is dominated by FFs (IEA, 2021) that were responsible for the dramatic growth of population and economic activity in the 20th century, particularly in coastal areas. It is also the primary cause of climate change and economic activity that has led to widespread environmental deterioration. The current, growth-centered neoclassical economic framework grew out of western colonialism and coincided with the rapid expansion of population and economy over the past century and a half but is poorly equipped to address global constraints and diminishing resource availability (Hall, 2017; Xu et al., 2020; Day et al., 2021). We address these issues in more detail and look to the future of coastal systems in the remainder of the article.

## Climate change impacts and coasts

Because of Holocene stability, climate has been hospitable for humans and agriculture for thousands of years. But now climate change is of significantly greater magnitude than during the Holocene and is dramatically impacting the earth system and global society. Coastal zones are at the forefront of areas threatened by climate change in terms of direct impacts due to warming of the atmosphere and oceans, accelerated sea level rise, larger and more intense tropical cyclones, extreme precipitation events, and changes in river discharge and especially given the intense development there. Other climate change forcings impact coastal areas indirectly through drought, water stress, wildfires, and melting polar sea ice, while others are more indirect, for example, drought and decreased precipitation in river basins affect freshwater delivery to coasts (Stott et al., 2004; Emanuel, 2005; Webster et al., 2005; Hoyos et al., 2006; FitzGerald et al., 2008; Pfeffer et al., 2008; Vermeer and Rahmstorf, 2009; Kaufmann et al., 2011; Min et al., 2011; Pall et al., 2011;

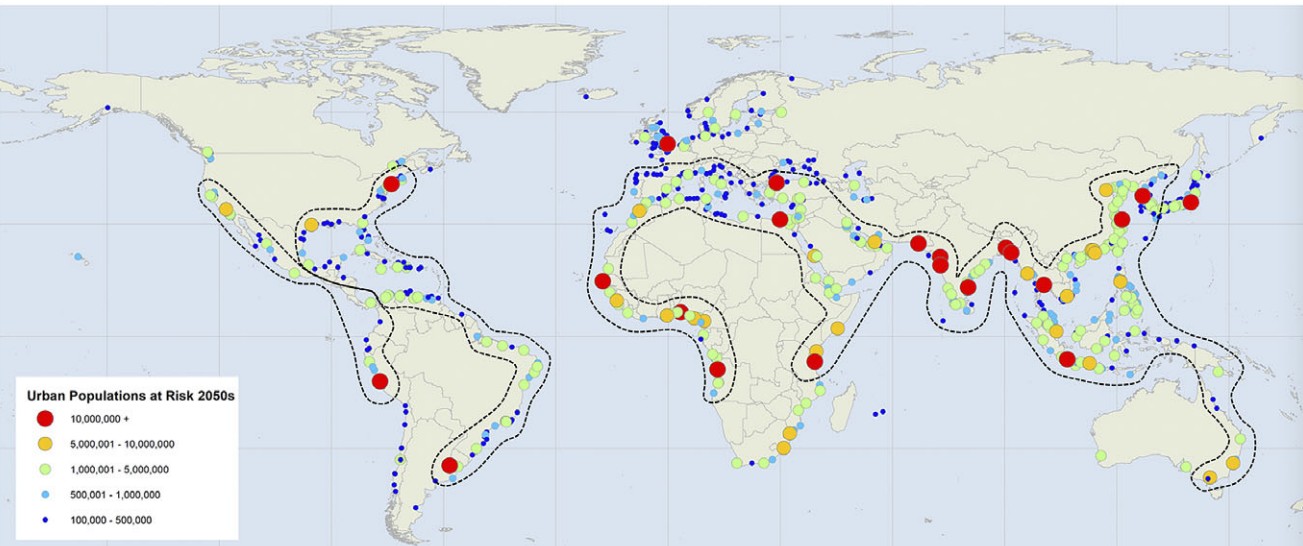

**Figure 1.** The current global distribution of coastal urban areas larger than 100,000 people. Many of the largest cities are in lower income, tropical to subtemperate coastal regions including tropical Africa, South and Southeast Asia, Central America, and the Caribbean (C40.org 2020). The coastal zones enclosed within the dotted lines area are projected to experience multiple climate stresses that will interact with social, political, and economic megatrends that will likely lead to diminished resilience and sustainability (Modified from C40.org 2020 by Day et al., 2021. Used with permission).

**Table 1.** Top 25 largest megacities of the world as of October 2022 (modified from United Nations, 2022)

| Rank | City | Country | Population (mil) | Coastal | Delta region | Human development index |
|------|------|---------|-----------------|---------|--------------|-------------------------|
| 1 | Tokyo | Japan | 37.39 | Yes | No | Very high |
| 2 | Delhi | India | 30.29 | No | No | Medium |
| 3 | Shanghai | China | 27.05 | Yes | Yes | High |
| 4 | Sao Paulo | Brazil | 22.04 | Yes | No | High |
| 5 | Mexico City | Mexico | 21.78 | No | No | High |
| 6 | Dhaka | Bangladesh | 21 | Yes | Yes | Low |
| 7 | Cairo | Egypt | 20.9 | Yes | Yes | High |
| 8 | Beijing | China | 20.46 | Yes | No | High |
| 9 | Mumbai | India | 20.41 | Yes | No | Medium |
| 10 | Osaka | Japan | 19.16 | Yes | Yes | Very high |
| 11 | New York | USA | 18.8 | Yes | No | Very high |
| 12 | Karachi | Pakistan | 16.09 | Yes | Yes | Low |
| 13 | Chongqing | China | 15.87 | No | No | High |
| 14 | Istanbul | Turkey | 15.19 | Yes | No | Very high |
| 15 | Buenos Aires | Argentina | 15.15 | Yes | No | Very high |
| 16 | Kolkata | India | 14.85 | Yes | No | Medium |
| 17 | Lagos | Nigeria | 14.36 | Yes | Yes | Low |
| 18 | Kinshasa | Congo | 14.34 | Yes | No | Medium |
| 19 | Manila | Philippines | 13.92 | Yes | No | Low |
| 20 | Tianjin | China | 13.58 | Yes | No | High |
| 21 | Rio de Janeiro | Brazil | 13.45 | Yes | No | High |
| 22 | Guangzhou | China | 13.3 | Yes | No | High |
| 23 | Lahore | Pakistan | 12.64 | No | No | Low |
| 24 | Moscow | Russia | 12.53 | No | No | Very high |
| 25 | Los Angeles | USA | 12.44 | Yes | No | Very high |

*Note:* Country Development Status is from the Human Development Index (https://hdr.undp.org/data-center/human-development-index#/indicies/HDI). Coastal is within 200 km of the ocean.

Schiermeier, 2011; IPCC, 2013; Horton et al., 2014; Mei et al., 2015; Deconto and Pollard, 2016; Kopp et al., 2016; Day et al., 2021).

Extreme heat is impacting large areas of the globe. Global temperatures have increased by a little more than 1 C since 1880 and are projected to increase by an additional 1.5–2.0 C or more by mid-century, coinciding with a disproportionate increase in the frequency, duration, and geographic extent of heat wave events (IPCC, 2022). For the last 6,000 years, human populations, crops, and livestock occupied a limited thermal window compared to global temperatures (Xu et al., 2020). Without a reduction in emissions, this will likely shift even more in the coming decades than was typical of the past 6,000 years. A third of humans may experience mean annual temperatures exceeding 29 C by 2070 in comparison to most living in the nearly optimal 11–15 C range for the past 6,000 years. Especially vulnerable are tropical zones, especially along tropical coasts, because these areas are already in the high end of the temperature range, leading to temperatures exceeding human thermoregulatory capacity. Thirty percent of the world's population is currently experiencing very high temperature and humidity conditions for part of the year, and that proportion will increase to between 48 and 74% by 2100 (Mora et al., 2017). Increasing mortality from high heat is inevitable, and this is

especially true in humid coastal cities where urban 'heat islands' exacerbate the problem. Increasing frequency and duration of extreme heat will inevitably require increased energy demand for cooling, leading to inequities in mitigating the effects of climate change (Hughes et al., 2023). Increasing heat poses particular challenges for food security in cities globally (Hammond et al., 2015) but especially in lower income tropical regions. Marine heat waves have increased in frequency and duration over time (Oliver et al., 2021), raising the question of ocean heat absorption capacity and increased abrupt die-offs of marine resources (e.g., https://onlinelibrary.wiley.com/doi/10.1111/gcb.16301). In just 16 years (from 2005 to 2021), heat stored by the earth system doubled with possible irreversibly effects to the climate envelope in which humans live (Loeb et al., 2021). As a result, a third of ice in Himalayan glaciers would be lost with 1.5 C warming and 50–65% at higher warming levels (Kraaijenbrink et al., 2017).

Oppenheimer et al. (2019) reported sea level rise estimates and projections of from 1986–2005 to 2081–2,100 (ranges in brackets) (0.43 m (0.29–0.59)-RPC2.6, 0.55 m (0.39–0.72)-RPC4.5, and 0.84 m (0.61–1.10)-RCP 8.5). Especially vulnerable are low-lying coastal regions including tropics and subtropics (see Figure 1) with large deltas and high population densities (Syvitski et al., 2009,

2022; Giosan et al., 2014; Day et al., 2016, 2019, 2021). Coastal flooding is projected to be even worse due to climate-induced tidal and storm events (Kirezci et al., 2022) The world's megacities are primarily in tropical coastal areas (see Table 1). The legacy of anthropogenic impacts to drainage basins of large deltas through localized reclamation, hydrologic alteration, and enhanced subsidence are decreasing the resiliency and sustainability of these areas (Giosan et al., 2014; Tessler et al., 2015; Day et al., 2016, 2019, 2020).

Warming of ocean surface waters has amplified the frequency and intensity of tropical storms including cyclones and surge events (Emanuel, 2005; Webster et al., 2005; Hoyos et al., 2006; Elsner et al., 2008; Bender et al., 2010; Grinsted et al., 2012; Mei et al., 2015; Bhatia et al., 2019). The rate that the strength of hurricanes declines after reaching land has decreased over time (Li and Chakraborty, 2020). Countries along the western Pacific and Indian Ocean, which contain many coastal megacities (see Figure 1), are among the most densely populated globally (Burger et al., 2017, 2019, 2022; Burger and Fristoe, 2018) and are threatened by tropical cyclones. The impacts are greater in low-income countries that lack the capacity for socioeconomic resiliency to respond to increasing natural disasters (e.g., Dyer, 2009; Tessler et al., 2015).

Thus, ongoing climate forcings will affect cities broadly in tropical to subtemperate regions (Figure 1). Because of the multiplicative effects in the lower latitudes, tropical cities will be most vulnerable to cascading effects. Climate change is already impacting the world's coastal megacities and major deltas (Tessler et al., 2015; Day et al., 2021). The energetic costs and climate impacts of cities in combination with global change forcings raise questions about if urbanization will become more difficult to maintain in the future (Burger et al., 2012; Day et al., 2021; Kummu et al., 2022). In this review, we address the intersection issues of coastal human–natural systems in the face of global change.

## Energy, the renewable transition, and the future of the world's coasts

A careful consideration of the future trajectory of societal and biophysical systems in coastal regions requires an understanding of the world's energy system over the past two centuries. The current status of the biosphere and global industrialized society, including coasts, is primarily the result of massive FF use over that time period. This resulted in exponential growth in agricultural production, human populations, and economies, and an overall improvement in the material wellbeing of many humans. This FF growth fueled urbanization and economic growth (Burger et al., 2019) and the growth of megacities, most of which are coastal (Day et al., 2021). Unfortunately, it also led to widespread deterioration of many natural systems and dramatic climate impacts. Coastal systems in both their human and biophysical aspects are at the forefront of climate impacts. FFs underwrote practically all growth. This is reflected in the strong statistical relationship between GDP energy consumption patterns (Figure 2; Warr and Ayres, 2010; Brown et al., 2011, 2014; Ayres and Voudouris, 2014; Hall, 2017; Smil, 2022). FF use also allowed much of the global population to live far from the places where most of the resources are produced including energy, food and agricultural products, industrial goods, and services consumed by urban populations (Burger et al., 2012). Marine transportation networks powered by FF deliver these goods to their destinations and are a central reason why so many people can and do live in coastal zones.

Even with rapid growth of renewables, FFs still dominate global primary energy use by humans, providing more than 80% of global primary energy and 75% of energy end use in 2020 (BP, 2021). Thus, FFs are still supporting most economic activity including renewable energy development, and also dealing with the economic impacts of climate change. Fossil fuels are also integral to maintaining human infrastructure and driving discretionary consumption spending. While the development of renewables is growing rapidly, they are largely adding to the total energy available and not substituting for fossil fuels FFs (IEA, 2017). Conventional oil production has reached a peak and is declining now for 38 of 45 oil producing nations and for 6 of 8 continents (Hallock et al., 2004; Mushalik, 2021; Laherrère et al., 2022).

Both energy use and GDP grew exponentially in the 20th century (Figure 2). By 2011, global energy use was about 500 quadrillion BTU (approximately 500 EJ or 100 billion barrels of oil equivalent; Figure 2). Of this, U.S. demand accounted for nearly

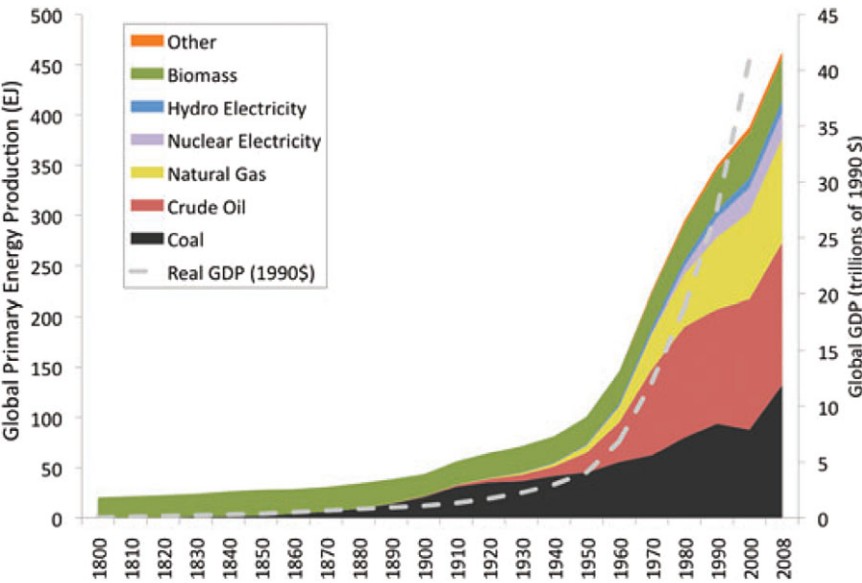

**Figure 2.** Growth of world energy production and GDP from 1830 to 2008. "Other" includes solar, wind, and other new renewables (Murphy and Hall, 2011, used with permission).

one-quarter. According to BP (2021), global primary energy consumption increased 2.7 times from 1970 (204 EJ) to that in 2020 (557 EJ). FFs provided 82% of primary energy in 2020. Oil is the most important FF, followed by coal and natural gas. All other primary energy sources accounted for ~18% in 2020: nuclear, hydropower, and biomass (wood, peat, and dung) were ~ 12–14% and photovoltaics and wind turbines about 4%.

Most projections for future energy use show FFs dominating though mid-century but then declining rapidly (Maggio and Cacciola, 2012; Mohr et al., 2015). The IEA (2021) projects that oil use will level off at about 104 mb/d by the mid-2030s and then decline slowly to 2050. Natural gas will grow to around 4,500 billion m$^3$ in 2030. Global coal demand will grow slightly until 2025 and then decline slowly to 2050. Maggio and Cacciola (2012) and Mohr et al. (2015) project that total FF production will peak around 2050 and decline thereafter. These projections are driven by depletion and decreasing net energy of FFs and do not assume replacement by renewables.

Laherrère et al. (2022) developed a global assessment of the quantity of oil that remains to be produced, and how much this might limit future global economic development. They first defined different categories of oil and related liquids. They showed that public-domain data for "proved" (1 P) oil reserves, both by-country and globally from the EIA and BP *Statistical Review*, were extremely suspect. Oil consultancy proved-plus-probable (2P) reserve data are generally backdated to the date of field discovery in attributing a field's estimated volume and are therefore more reliable (see Laherrère et al., 2022). They concluded that the amount of remaining oil is probably overstated by 300 Gb in the Middle East and 100 Gb for the Former Soviet Union (FSU). Ultimately, the best statistic to assess how much oil is left to produce in a region is recoverable resource (URR) after production to date is subtracted. Laherrère et al. estimated the global URR for four aggregate classes of oil using Hubbert linearization (HL). Their results ranged from 2,500 Gb for

conventional oil to 5,000 Gb for 'all-liquids'. When oil produced to date is subtracted, estimates of global conventional oil reserves are only about half the EIA estimate. They then estimated the expected dates of maximum *resource-limited* production globally of the different classes of oil. The time of maximum production ranges from 2019 for conventional oil (the high-quality product that civilization was built on) to about 2040 for 'all-liquids'. In other words, the world is beginning to see the gradual end of freely available oil. We recognize that "peak oil" has been predicted before, but now it seems to be happening.

### Projections of future energy use through the renewable transition

In this section, we report several contrasting views on the resources and needs to move to a renewable world. The contrasts aside, the need to transition is inescapable, and the resources to do so remain limited.

There have been a number of projections of future energy use where renewables are included. Among them is the IEA (2013) that reported FFs provided about 83% of total primary energy in 2011 (Figure 3). Wind and solar were less than 1%. Coal and natural gas were most important for electrical generation with smaller amounts from hydro and nuclear. Residential, industrial, and transport, respectively, represented 32%, 39%, and 28% of energy end use. About 10% of primary energy inputs, mainly oil, were for non-energy uses such as petrochemicals.

Four years later, the IEA (2017) reported electricity accounted for only about 18% and 22% of world energy end use in 2011 and 2015, respectively. But in a renewable future, all FFs must be replaced with renewables and nuclear. Replacing present FF-generated electricity by wind and solar using the existing grid is possible but extremely challenging. However, replacing all FF

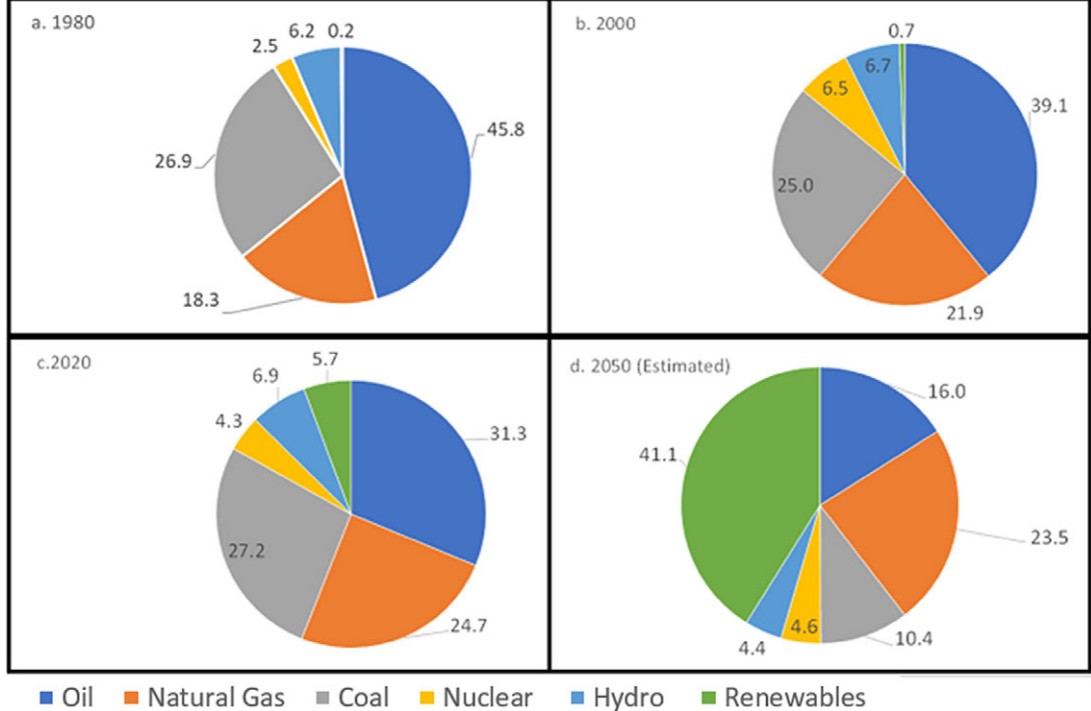

**Figure 3.** World Primary Energy Consumption by Percent Fuel Type from (a) 1980, (b) 2000, (c) 2020 (Source: Modified from BP Statistical Review of World Energy 2021), and (d) projection to 2050. The total energy consumption is 279.4 EJ, 394.5 EJ, 557.1 EJ, and 589.7 EJ for the corresponding years, respectively (Source: calculated from BP Statistical Review of World Energy 2021 for 1980, 2000, 2020, and DNV, Energy Transition Outlook, 2021 for 2050 – estimated).

uses with electricity is vastly more difficult and likely impossible because of the greater flexibility of FF energy (Day et al., 2018).

The IPCC (2022) projected energy use for two net-zero $CO_2$ emissions scenarios. By 2045, FFs provide 211 EJ and wind and solar 210 EJ. BP (2021) reported and projected FF for the years 1980, 2000, 2020, and 2050. Their Figure 3 indicates that FFs provided 91%, 86%, and 83% of global primary energy in those years and are projected to supply about 50% by 2050.

At the same time, renewables supplied 0.2%, 0.7%, and 5.7% with a projected 41.1% by 2050. Thus, there has to be an enormous decline in fossil fuels and increase in renewables for this scenario to occur.

For humanity to move toward a sustainable trajectory, an energy transition must occur; however, economic growth requires increasing energy availability, and this will be difficult (e.g., Hall, 2017; Day et al., 2018). Whether society can rapidly transition toward a sustainable pathway and meet the demands for provisioning of basic human needs is an open question.

Maggio and Cacciola (2012) provide rate of growth comparisons that show us how rapidly industrialization was occurring: the number of years it took fossil fuels to grow by an order of magnitude to 2010 levels was total FF 77, oil 64, coal 110, and natural gas 65. By contrast, some projections for the renewable transition are much more rapid. For example, Sgouridis et al. (2016) reported that to replace FFs by 2050, wind and solar would have to grow from <1 Gboe in 2010 (giga barrels of oil equivalent) in 2010 to 102 Gboe by 2050, a factor of 170. By 2019, installed wind and solar represented about 8% of total electricity usage. However, electricity currently represents about 18% of final global end use (Smil, 2022), most of which is generated by burning FFs and, by contrast, only about 2% of current total primary energy end use.

Another issue is that if society is to move rapidly energy consumption dominated by FFs to renewables, which generally have a lower energy return on investments (EROIs) to meet climate targets, society would have to allocate substantially more of its GDP to investing in new infrastructure for energy storage, renewable energy sources, and electric power grids. This allocation would reduce EROIs, compete with other economic drivers, and reduce consumption (Hall et al., 2009; Day et al., 2018; Capellan-Perez et al., 2022). Jacobson et al. (2015a, 2015b) estimated that the transition to an entirely renewable energy would require the allocation of 5% of U.S. GDP by 2040. In addition, decommissioning the existing FF infrastructure would require significant quantities of FFs. This would severely limit discretionary investments that facilitate economic growth and reduce EROI (Lambert et al., 2014; Hall, 2017).

Coastal systems are especially vulnerable to changes in the energy system since they are disproportionally impacted by climate change and are areas of high population density and economic growth. The findings of Laherrère et al., 2022 have profound implications for the renewable transition and are deeply seated in society's present dependence on FFs. Currently, FFs support more than 80% of the work of society (Figure 3). This includes infrastructure maintenance and consumption as well as investments in obtaining more energy (including the development of renewables). FFs also pay for dealing with damage due to impacts from climate changes and ecosystem degradation, which in turn are mostly caused by FF use.

## Economics: The failure of neoclassical and the promise of biophysical economics

How would we plan for, design, or project the ways that humans live and work in a future world where humans are becoming more urbanized and concentrated in increasingly fragile coastal areas? How, especially, could we design such a future that would be more sustainable in a world challenged by what we might consider a "perfect storm" of environmental deterioration, climate change, resource scarcity, and the coming end of the fossil fuel era? We believe that conventional economics is seriously inadequate for addressing the needed changes because it is inconsistent with the basic laws of the natural sciences, is based upon an unrealistic set of assumptions about human behavior and market structure, and does not effectively incorporate any of the new challenges that we identified earlier (see e.g., Hall et al., 2001; Hall, 2017). Although it has gone largely unheeded, there is a long history of critiques of the essentially absurd assumptions of NCE that violate many laws of physics (e.g., Boulding, 1968; Georgescu-Roegen, 1971; Cleveland et al., 1984; Hall et al., 2001; Hall, 2017). Herman Daly's many works on steady-state economics, sustainability, and ecological economics are a scholarly and convincing critique of NCE (Daly, 1977, 2007; Daly and Cobb, 1989; Daly and Farley, 2004; Daly, 2014). The shortcomings of NCE manifest most profoundly in coastal areas where one of the most dire impacts of modern industrial society, that of climate change, are being felt most strongly.

The neoclassical conception of the rational, maximizing, self-regarding consumer in a world where perfect information is represented in present day prices is incompatible with future sustainability. If the neoclassical premise that human beings are insatiable in their desires for greater happiness, achieved by means of ever-increasing material consumption is correct, the chances of arresting climate change, resource depletion, and the transcendence of even more planetary boundaries are negligible. Yet this behavior is often prominent among the affluent in coastal cities where people flock to the greater incomes and employment, greater array of consumer goods, and higher energy lifestyles of the seacoast.

The world in which we live in is not characterized by the neoclassical vision of atomistic perfectly competitive firms that are unable to control, or even influence, prices; have neither technological advantage nor unique products; and are willing to accept no economic profit in the long term. Yet according to NCE, such an abstract and unrealistic economy is the only one that can produce efficiency and equity in the long run. Rather the actually existing economy is one dominated by large-scale concentrated corporations, or oligopolies, in sectors such as manufacturing, finance, agribusiness, and electronic communication. Highly concentrated corporations can maximize profits in the long run by controlling and externalizing costs and expanding their market shares. But when forms of market power (imperfect competition) are considered, long-run solutions are neither efficient nor equitable. Rather they are characterized by inefficiency, exploitation, periodic depressions, and an unequal distribution of incomes. An oligopolized market structure not only produces the perpetual need to accumulate, as implied in the need to increase market share, but also increased inequality and environmental destruction as a consequence of seeking to control costs.

However, neoclassical economics treats environmental problems of all sorts as external to the main analysis, and if it does think about the environment, it is only by (barely) moving the environmental problems into the market system as "externalities", subject to cost–benefit analysis to ensure that the cost of remediation or adaptation is not too high for the present generation.

There is a need for a new economic paradigm that is consistent with the constraints imposed by earth systems and yet integrates the

limits found in nature with the limits found in the internal dynamics of a globalized concentrated market structure dominated by finance and not the entrepreneurial firm. One alternative approach to economics starts with natural science and heterodox political economy and builds on them.

This integrative framework, which we call *biophysical economics*, would address these problems by integrating the material and energy needs and possibilities into the initial assumptions and analyses of economic theory. A crucial starting point of biophysical economics is to conceptualize the economy as a flow of materials and energy, and not just a flow of abstract value, or money. While money can theoretically expand forever, materials and energy are limited by the finite nature of earth systems, and biophysical limits to growth are real on a finite planet. As we deplete our earthly sources and fill our sinks with societal effluents, the limits inch closer with every passing day, taking the form of increasing atmospheric carbon dioxide, depletion of crucial resources, including fossil fuels, ocean acidification, and increasing pollution, as we approach and exceed, our planetary boundaries (Rockstrom et al., 2009). Many scientific analyses are well aware of these physical and biological phenomena.

What makes biophysical economics unique is the integration of limits to growth found in nature with those found in the internal dynamics of the concentrated economy itself. While in the past, the economy needed to grow to remain viable, strong internal tendencies produce periodic recessions as well as long-term stagnation, or slow growth. This stagnation appears to be increasing, and the average growth rate for the United States during the 1940s was nearly 6% per year, but after domestic oil peaked in the continental United States, growth rates began to fall over time (Hall, 2017). The average growth rate during the 2010s was only 1.9%, and the first quarter of 2023 produced a growth rate of only 1.1%. While in the past when the large economic surplus was not consumed or invested, it had to be wasted in endeavors such as advertising or in an energy grid based upon geographically concentrated electricity generation and long-distance transmission. But in the future, neither conspicuous consumption nor copious waste is a viable strategy for a sustainable economy, especially a coastal economy. All the surplus, and probably more, will be absorbed by attempting to maintain increasingly vulnerable infrastructure. Most citizens will see the result as inflation, which will make polities increasingly difficult to govern just as they need good governance.

In short, NCE is no longer up to the task of guiding society and the economy. There is a need for a rapid transition to biophysical economics, which respects the laws of nature and the functioning of the biosphere, and analyzes the economy as it actually exists. A viable theory must address the material and energy requirements, as BPE does, and not simply the growth in terms of money to devise a solution to a society whose growth now pressures our planetary boundaries. Coastal areas are emblematic of these problems because 50% of the world population lives within 200 km of a coast, economic activity is increasingly concentrated in coastal areas, especially tropical ones, and these areas are in the forefront of the impacts of climate change and ecosystem deterioration. Because coastal systems are among the most productive ecosystems on earth, they provide vast ecosystem goods and services to society. Much of the global economy is focused on coastal areas, and most of the world's trade passes through coastal ports. However, these areas are among the most threatened globally. These factors must be addressed by economic analyses. The framework of biophysical economics encompasses these realities.

## Provisioning of coastal social and economic systems

What does it take to support global coastal populations? Certainly, maintaining local coastal populations and infrastructure is critically important because the majority of global population is coastal. But combined global forcings will impact and is already impacting people and the ability to maintain infrastructure. Often resources, especially increasingly expensive energy, that are used to maintain coastal populations are sourced from long distances that are mostly non-coastal and growing global stresses will make this increasingly difficult. Coastal economies in poorer countries have fewer resources to address impacts on infrastructure and flows of energy and materials necessary to maintain coastal populations and economies and are less able to afford costs of dealing with climate change and environmental degradation (Tessler et al., 2015; Day et al., 2016, 2018, 2021).

The global energy system is critical and central to sustaining human populations and economies. The world is moving toward a transition to renewable energy, mainly wind and solar, but total world primary energy sources are still overwhelmingly dominated by fossil fuels. Even though renewable energy growth has been dramatic, new renewables still supply a small percentage of total world energy use. Renewables have not displaced fossil fuel use because of the continued growth in total global energy consumption. LaHerrère et al. (2022) showed that total petroleum resources that will ultimately be produced are only half of what was previously thought to be available. Currently, fossil fuels are required for almost all renewable development as well as all other work in society, including the rebuilding of the many coastal regions devastated by hurricanes and flooding. There will be increasing competition for fossil fuels, which underwrites economic activity that supports societal investments (energy acquisition, infrastructure maintenance, pensions and health care, discretionary spending for such things as parks, arts, and museums) and consumption (staples and discretionary spending). High levels of discretionary spending are what characterizes richer societies. An important question is can they be sustained as net energy resources are diminished (e.g., Hall et al., 2008)?

FFs are sourced from a globally distributed array of coal, and oil and natural gas fields (Figures 4 and 5). FFs have to be extracted from these fields, processed, and shipped to where they will be used. Because the ultimately recoverable petroleum reserves are only about half of what was projected by the IEA, there will be increasing shortages of petroleum in the relatively near future and the net energy yield (EROI) will be less. These increased costs will weigh heavily on the economy, especially in coastal regions in developing countries. This suggests that the renewable transition will be very difficult, if not impossible, if the future economy is expected to look like the present.

Much of the food to feed growing coastal populations is sourced from a globalized food system spread over many parts of the earth. They are generally not near where coastal people live (Figure 6). Fouberg et al. (2015) list thirteen different food-producing systems. Seven of these are traded to a significant extent in the globalized industrial food system: dairying (1), fruit, truck, and specialized crops (2), mixed livestock and crops (3), commercial grain farming (4), Mediterranean agriculture (6), plantation agriculture, mainly tropical (7), and livestock ranching (12). Six of the food systems are

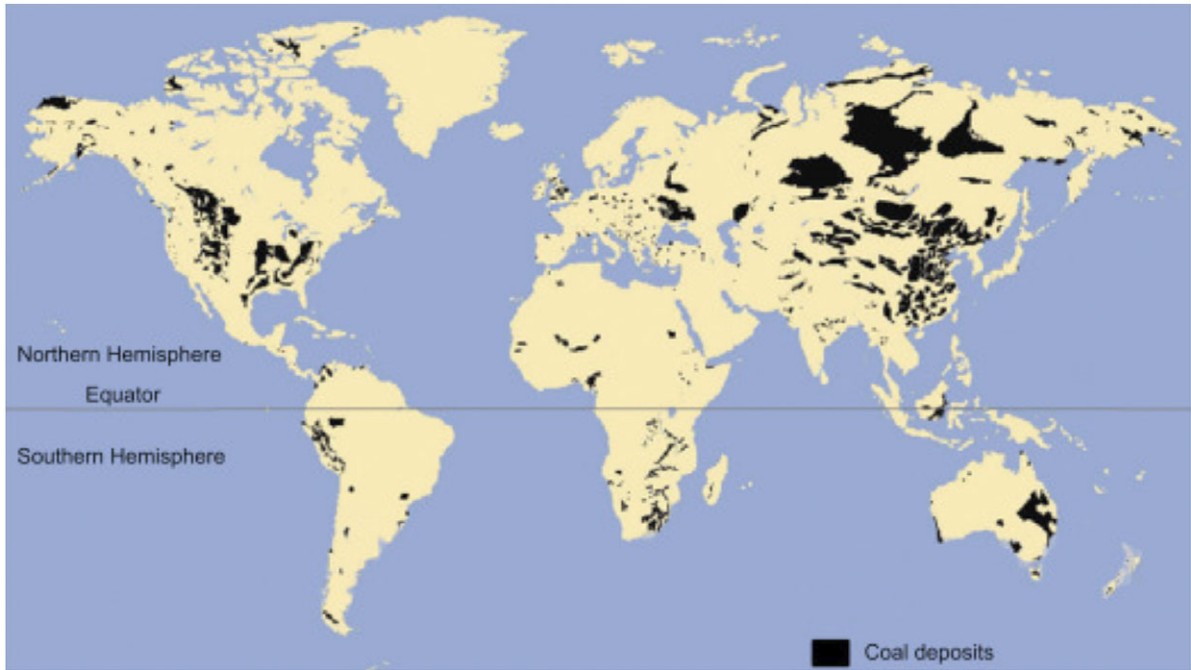

**Figure 4.** Major coal fields of the world (Suárez-Ruiz et al., 2019). Most are far from coasts. Used with permission.

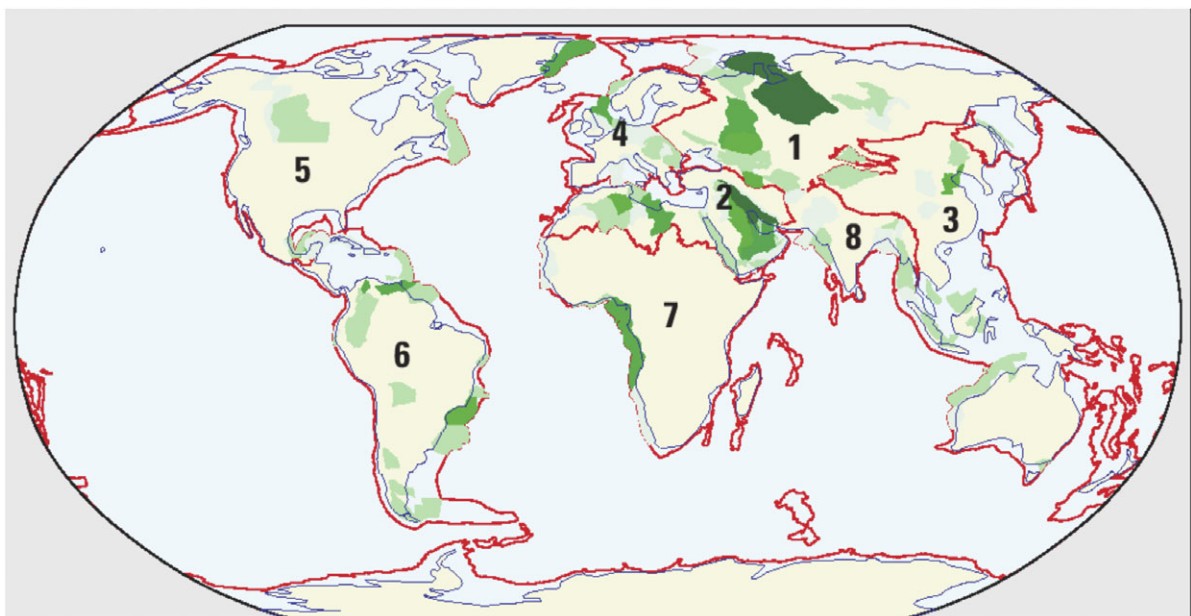

**Figure 5.** Major oil and gas fields of the world. Oil endowment (cumulative production plus remaining reserves and undiscovered resources) for provinces assessed (from USGS, 2003). Darker green indicates more resources. Measured regions include (1) the former Soviet Union, (2) the Middle East and North Africa, (3) Asia-Pacific, (4) Europe, (5) North America, (6) Central and South America, (7) sub-Saharan Africa and Antarctica, and (8) South Asia (USGS, 2003). Alaska and the U.S. Gulf coast are not shown here. Red lines indicate the location of the edge of the continental shelf. Used with permission.

primarily subsistence food production: subsistence crop and live-stock farming (5), intensive subsistence farming, chiefly rice (8), intensive subsistence farming, chiefly wheat and other crops (9), rudimentary sedentary cultivation (10), shifting cultivation (11), and nomadic and seminomadic herding (13). This latter group is traded to a lesser extent in the global food system or not at all.

Producing the food traded in the global food system is highly energy intensive. The average item of food in the United States travels an average of 1,500 miles from where it is produced to where it is consumed. From the farm field to the plate, the global food system requires 15–30% of total primary energy (Marshall and Brockway, 2020; Schramski et al., 2020). Country-level food self-sufficiency has declined over the last four decades (Schramski et al., 2019), and, as we have shown, many of these countries are in tropical coastal zones. Climate change and environmental degradation are strongly impacting food production (e.g., Xu et al., 2020). It is certain that increasing energy prices and resource shortages will severely impact the food system.

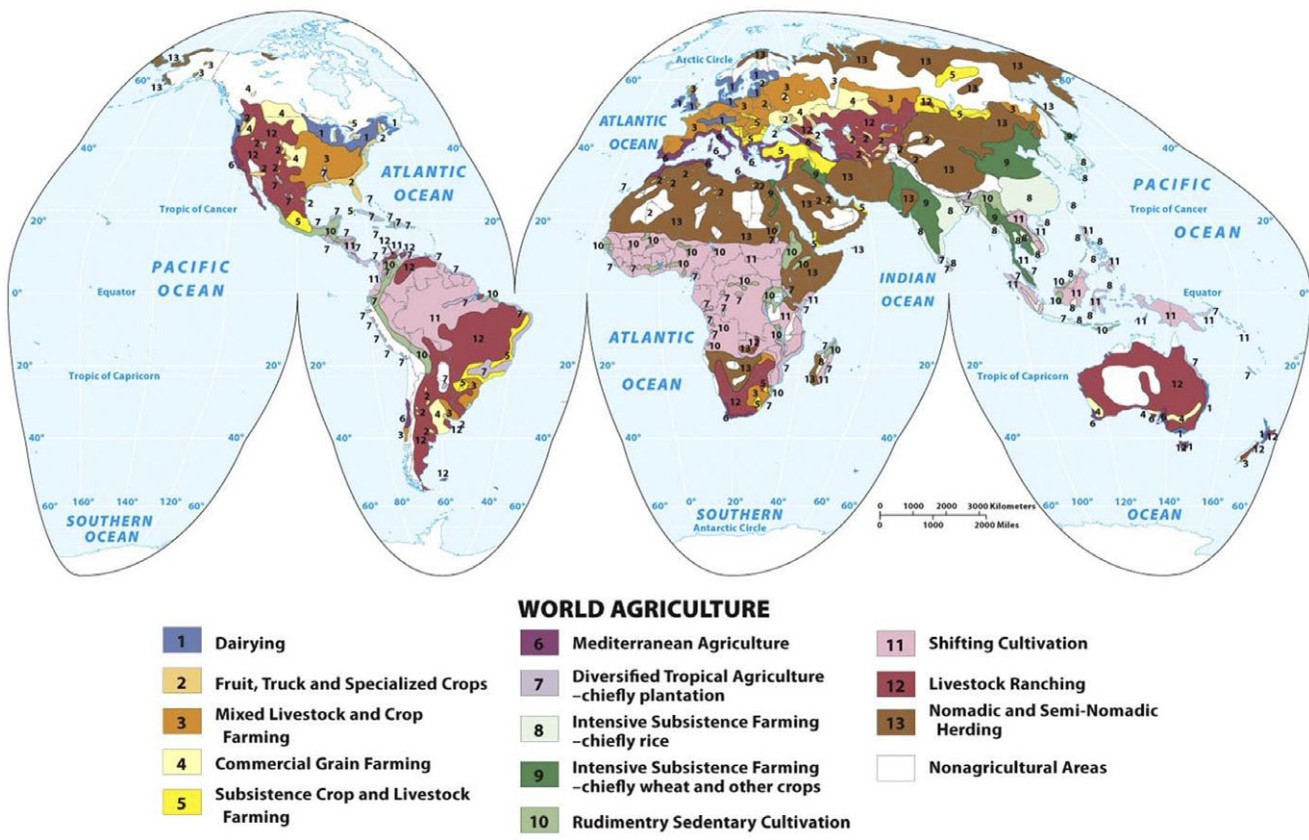

**WORLD AGRICULTURE**

| | | |
|---|---|---|
| **1** Dairying | **6** Mediterranean Agriculture | **11** Shifting Cultivation |
| **2** Fruit, Truck and Specialized Crops | **7** Diversified Tropical Agriculture –chiefly plantation | **12** Livestock Ranching |
| **3** Mixed Livestock and Crop Farming | **8** Intensive Subsistence Farming –chiefly rice | **13** Nomadic and Semi-Nomadic Herding |
| **4** Commercial Grain Farming | **9** Intensive Subsistence Farming –chiefly wheat and other crops | Nonagricultural Areas |
| **5** Subsistence Crop and Livestock Farming | **10** Rudimentry Sedentary Cultivation | |

Figure 11.18

**Figure 6.** Major food-producing areas of the world. Used with permission.

Likewise, the globalized trade system that provides the goods and services that maintains human society is widely distributed and consumes enormous quantities of goods and energy and will also be impacted by growing energy scarcity and climate change. The most important exchange points in the global trade system are coastal cities, and these are threatened by several ongoing, climate change processes. For example, the ports of Shanghai, New Orleans and the lower Mississippi River, and Rotterdam are in deltas with large areas below sea level. The former two are in the tropical cyclone belt. The port of Los Angeles is in a region being consumed by drought and fire. The whole world trading system is, of course, underwritten by cheap fossil fuels. Jackson and Jensen (2022) argue that the future will be characterized by fewer people consuming fewer resources. This has dire implications for the globalized industrial society, especially coastal systems that are often far removed from major food production areas (Kummu et al., 2022).

### The world's coasts: Shifting baselines and diminishing sustainability – Summary and conclusions

In this section, we summarize the environmental baseline under which coastal systems and societies developed during the Holocene and then describe how this baseline has changed and the implications for future sustainability.

Compared to the late Pleistocene, the Holocene climate of the past ten thousand years was relatively stable. Global average temperature was about 5 C warmer than that during the glacial period.

This resulted in a wetter climate and significantly higher terrestrial and marine global net primary productivity (NPP). This was especially pronounced over shallow shelfs as rising sea levels flooded the continental margins. Since stabilization about six thousand years ago, sea level has been relatively stable, varying by only a few meters after rising about 130 m. This was the period of Holocene delta building, and coastal margin productivity increased after sea level stabilization (Day et al. 2007a, 2007b, 2012a, 2012b). In general, coastal margins experienced much higher levels of freshwater and material inputs from drainage basins. Areas that were covered by ice sheets experienced isostatic rebound. Large deltas grew rapidly after level stabilization but experienced subsidence due to sediment loading, which was offset by deposition of sediments from continental sediment input as well as biogenic substrate formation.

The human population grew slowly from the beginning of the Holocene to about one billion by 1800. Beginning about 12,000 years ago, agriculture spread rapidly and many people settled in small villages. Between 7,000 and 5,000 years ago, the first urban areas appeared, primarily in coastal margins, and often associated with deltas as in Mesopotamia, Pakistan, China, Mesoamerica, and Peru. The massive increase in coastal margin productivity coincided with urbanization that was typified by increasing social complexity and inequality and monumental architecture that included large public works (Day et al., 2007b, 2012b; Gunn et al., 2019). From the beginning of agriculture to the beginning of the industrial revolution in the 18th century, humans lived sustainably from the ecosystem goods and services provided by the biogeosphere. Societies waxed

and waned due to factors such as climate variability and ecosystem degradation (Tainter, 1988, 2005; Diamond, 2005; Chew, 2007), but overall population increased, and humans spread over the habitable land surface of the earth. Although human activity significantly impacted the earth system locally, it never reached levels that affected global sustainability until the 20th century.

This changed with the advent of the industrial revolution powered by FFs. Industrialization originally was based on water wheels, windmills, and burning wood, all of which can be considered as ecosystem services. Without FFs, the industrial revolution would have quickly become limited by energy scarcity. It was FFs that supported the dramatic exponential growth of energy use, the economy, population, and many other processes in the 20th century. This dramatic growth has been termed the great acceleration and led to a new geologic epoch called the Anthropocene (Steffen et al., 2015; Syvitski et al., 2020). Most economic activity and population are now concentrated in coastal areas. Abundant, cheap fossil fuels played a central role in underwriting all these trends. Continuation of this growth seems impossible in a future of ever-increasing resource constraints, climate impacts, population growth, and environmental deterioration. This is particularly true for coastal areas, which are among the social and natural systems most threatened by climate change. Past growth rates were self-limiting and relatively short-lived compared to the long history of earth and humanity.

The industrial revolution has seen both unprecedented human numbers and wealth but also dramatic environmental deterioration and increasingly catastrophic climate change. Both natural and human systems in the coastal zone are affected by climate change and increased subsidence due to fluid withdrawal and reclamation. The finding that the ultimately recoverable petroleum resources are only half of what has been projected, meaning that it is unlikely that there will beadequate energy resources to support the renewable transition, meeting sustainability goals, infrastructure mainten-ance, and pay for the cost of climate impacts, while providing for much of discretionary spending. Unfortunately, there will likely be sufficient FFs burned to lead to substantial climate impacts in the coastal zone. Further population and economic growth will lead to increased pressure on resources and greater climate impacts and environmental deterioration. It seems inevitable that the great acceleration will slow and reverse, even while human pressure will lead to further exceedance of the carrying capacity of the earth.

Most population growth is currently occurring in developing countries in the tropical zone. The net result of the "exponential" century is that the great source and sink functions of the biosphere have been greatly diminished. Climate change impacts have grown so dramatically that many investigators have concluded that cata-strophic climate impacts are irreversible and coastal systems are at the forefront of climate change. Careful planning, cooperation, and likely Draconian legislation will be necessary to minimize the social damage of coming changes and not blind faith in unregulated markets. It seems inevitable that past patterns of energy use, resource consumption, and economic growth cannot be sustained. Neoclassical economics is conceptually and practically unable to deal in a meaningful way with this suite of problems, and a new economic system based on the laws of nature is necessary to guide humanity through the coming transition.

In summary, there are a series of overlapping issues that are likely to coalesce in a perfect storm of impacts on humanity as it gathers increasingly in an increasing impacted and fragile coastal zone. More and more energy will be required to address impacts of climate change and to compensate for and rebuild the human and natural infrastruc-ture located there. Food will be grown at locations far from human

populations, probably with increasing use of fossil fuels, while coastal food production will be increasingly stressed. Transitioning away from fossil fuels, necessitated by depletion and to reduce climate impacts, will be energy intensive and may use much of the remaining fossil fuels. Humans are psychologically not prepared for this as the dominant economic systems tell us that the material desires of human choices are what are most important and are infinitely possible to fulfill. All of this is likely to be felt most powerfully in the coastal zone, and society is largely unprepared for dealing with it.

**Open peer review.** To view the open peer review materials for this article, please visit http://doi.org/10.1017/cft.2023.23.

**Data availability statement.** No original data were used in this review article.

**Acknowledgements.** We thank 3 reviewers and Dr. Tom Spencer for thoughtful comments. Jessica Stephens helped with the production and sub-mittal of the paper.

**Author contribution.** All authors contributed to the conceptualization, writ-ing, and review of the manuscript.

**Financial support.** This work did not have any direct financial grant support.

**Competing interest.** The authors declare none.

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
