## [Reviewer Report]

I suggest that the authors carefully read the manuscript, as I found numerous dropped conjunctions and prepositions. Often paragraphs end with the topic sentences rather than beginning with them. Many paragraphs seem to have no topic sentence and are compilations of facts and figures strung together.

The following are some specific examples and suggestions.

Awkward phrase…meaning unclear. “This has profound implications for the renewable transition as FFs underwrite more than 80% of societal, including the transition to a renewable economy.” 

Awkward phrase…During the Holocene, the earth warmed and became wetter became more productive. 

Awkward…“Even with rapid renewables growth” how about, Even with the rabid growth of renewable energy…?

Awkward… “Thus, FFs are still supporting most economic activity including renewable energy development, dealing with the economic impacts of climate change, as well as maintenance of infrastructure, and consumption of staples and discretionary spending”….. Do FFs deal with economic impacts?

Figure 2 is a bit old…doesn’t show the economic/energetic downturn of the great recession (or recent Covid downturn) a fact that might help to make your point about the coupling of economy and energy. Aa more up to date graph would be useful.

Beginning on page 6 – LaHerrere et al. 2022- seems like too much detail. A summary of key finding would be enough.

“There have been a number of projections [of] future energy use where renewables are included. IEA (2013) reported that FFs provided about 83% of total primary energy in 2011 (Figure 3). “ There is no 2011 in Figure 3. And the data given in Figure 3 is from BP, not IEA.

“Rapidly moving to lower Energy Return on Investment (EROI) …” most interesting statement. It would be important to provide some background on these lower EROIs. This is the first the reader is introduced to such an important concept.

Awkward phraseology, meaning not clear…”The transition to renewable energy would require the allocation of 5% of U.S. 2016 GDP to renewable energy between 2025-2040 (Jacobson et al. 2015a,b), in addition to decommissioning FF infrastructure and FFs to provide remaining net energy requirements. “

The first sentence in section titled “Economics: The failure of NCE… “How, especially, could we design such a future that would be more sustainable in a world challenged by environmental deterioration, climate change, resource scarcity, and the coming end of the fossil fuel era?” … suggests that you are about to provide some insights on planning for the future…However instead, this section of the paper delves into a critique of NCE and a suggestion for a rapid transition to BPE. While I agree, a new economic is needed, it would be nice for other readers to understand a bit more how the transition to BPE would help with the major problems you have done such a good job elucidating. 

“These increased costs will weigh heavily on the economy, especially in coastal regions in developing countries.” ….Why especially in coastal regions?

Unclear meaning….There will be sufficient FFs used to lead to all but the worst climate impacts. 

The last sentence in the abstract suggests that ultimately after numerous pages of rehashing the present and future problems of global energy estimates, neoclassical economics and climate impacts on the coastal zone, that some suggestions, or at the very least some thoughts on potential scenarios for “careful planning and cooperation” that might “minimize impacts of these changes” might be offered. When I reached the final paragraph, I was surprised, and a bit let down. “It seems inevitable that past patterns of energy use, resource consumption, and economic growth cannot be sustained”….and so?

---

## [Reviewer Report]

Excellent review. Ready to publish as is with minor grammar corrections. However, in my opinion the analysis of neoclassical economics should lead and the review be restructured based on this lead idea.

---

## [Reviewer Report]

This is a very interesting paper that makes a complete review of major issues related climate change effects on coastal zones.

I consider that the last paragraph are the conclusions of the paper and may be a subtitle for this section.

---

## [Editor Report]

There are three reviews here but only Reviewer 1 provides a thorough – and useful - review. (I comment on the restructuring around NCE suggested by Reviewer 2 towards the end of this response - where I also cover the Reviewer 3 comment). I agree with the introductory comments of Reviewer 1 and that the manuscript needs some re-structuring of sentences in places. There is also a tendency to be rather too list-like in places, with some over-referencing. Are 17 references really needed at the bottom of page 3 / top of page 4? In summary, there is a really good review paper here but it tends to get lost in long general arguments (the arguments in themselves are good…) and a lack of coastal focus. In particular the end of the review is disappointing. I know that it is easier to identify how we have got to where we are but I do think that there needs to be rather more on where we could go in the near-future, based around what has gone before in the review. This addition, in my view, is a critical need for manuscript acceptance – and pushes the changes asked for to ‘major revision’. I concur with the final comments of Reviewer 1 here. 

Under ‘Coastal Climate Change Impacts’ there is no need to repeat the benign Holocene argument as this is covered in the Introduction. It is important here to concentrate on coasts and so I do wonder if the paragraph beginning on line 3 on page 4 (22 lines) might actually be condensed into a few lines that could go into the page 3 text on climate impacts. The megacities arguments in the next paragraph are repetition from page 2. The last paragraph of this section on page 5 seems repetitive too. So I think there is a bit of work to do in sharpening – and shortening - pages 2 to 5. 

Under ‘Energy, The Renewable Transition, and the future of the world’s coast’, Reviewer 1 highlights some awkward phraseology and asks if an updated Figure 2 could be provided. Some work is also needed on Figure 3 and the related text. Again, I think it is important here to keep to general principles and not to go off into too much detail – a point made by Reviewer 1 for the bottom of page 6 to the start of the next sub-section. Reviewer 1 also asks for more on the EROI which seems reasonable. As none of pages 6, 7 and 8 have a coastal focus it is too long as it stands. At lines 6-9 on page 9 is a really perceptive statement about the place of coasts in this debate – could that not be developed further in a shorter overall section? 

‘Economics: The Failure of Neoclassical and the Promise of Biophysical’. Reviewer 1 has some useful things to say here. I agree that his potential restructuring has merit- ‘how the transition to BPE would help with the major problems you have done such a good job elucidating’. There is a lot of text here before we get to BPE (pages 9 to 12) and sometimes I feel it covers the same ground in multiple ways. I quite understand that it is valuable – essential - to bring Howard Daly’s scholarship to this argument but I am less convinced that we have to have the broad historical view from Roy Harrod and Joan Robinson to Thomas Piketty. Could this material not be condensed down to a series of points, made only once? And why is the BPE section not referenced? 

‘Provisioning of coastal social and economic systems’ seems to start by going back to the LaHerrere arguments on page 6 (and again on page 9). Could you not just refer back here and concentrate this section on food supply and trade (pages 14, 15)? 

‘The World’s Coasts: Shifting Baselines and Diminishing Sustainability’. The first paragraph on page 16 is not needed as the argument really starts with the second paragraph. Or would it be better to move energy into this section, and the renewables transition down to end this history of the human occupancy and use of the coast (which would give the structure suggested by Reviewer 2)? The final paragraph on page 17 really has been said before (I think Reviewer 3 is asking for this paragraph to be formally recognised as a Conclusion). It is then a bit of a shock to suddenly get to ‘Literature cited’. The sense is that there is a missing section on possible coastal futures underpinned by ‘good’ economics and sensible resource use.

---

## [Reviewer Report]

Overall, the text needs more subheadings. As it is now, the text jumps from one subtopic to another and would benefit from subheadings. It would also benefit from careful editing to make sure paragraphs are constructed properly with topic sentences followed by a few sentences to further develop or support and ending with a summary/conclusion. Many paragraphs are upended by ending with a sentence about coastal systems that should be the first or topic sentence.

There is a lot of repetition. A good editing to remove it is recommended.

The section on economics that begins on page 9 shifts to 1st person plural following nine pages in the passive voice. And then the 1st person is largely dropped on page 10. While not a critical flaw, one wonders why suddenly it is important to make the shift and then to drop it again.

The discussion of neoclassical economics beginning on page 10, while interesting, is overly detailed and a bit wordy. It seems much of this is more appropriate for a textbook on biophysical economics and less for an article in Coastal Futures. With a little elaboration (an additional sentence or two) the bulleted text on page 9 could take the place of these 2 pages.

There are numerous occasions throughout the manuscript where an equivalency between money and energy are implied, in doing so it weakens the manuscript and serves to confuse the reader who may be uninitiated to BPE. Early in the manuscript money is defined as “abstract value” but as the manuscript progresses, it becomes increasingly difficult to tell if the authors are talking about investing abstract value or energy or GDP (another abstract notion of value). This is a fundamental flaw and needs to be addressed. Careful editing to remove them is warranted.

On several occasions an equivalency between less energy and increased inflation or investments in energy infrastructure and inflation are made. While some readers may be able to make the jump from less energy to increased inflation, many will not make that connection. More explanation is needed. Why does less energy drive inflation? Less energy in and of itself does not cause inflation, however, if the money supply is increased to offset the slowdown in the economy caused by less energy, inflation can occur. Investing money in energy infrastructure or environmental mitigation does not necessarily cause inflation…a careful explanation is needed.

I have numerous corrections and comments in a PDF of the manuscript which should be sent to the authors.

---

## [Editor Report]

The reviews on the revision range from ‘accept’ to ‘major revision’. Having re-read the manuscript my assessment lies between these two decisions. I am also mindful of the fact that this paper has been in review for a long time and that you have made considerable changes already. It just needs one final push to get the ms into an acceptable shape now. The section on neoclassical economics and BPE (from line 34, page 9 to line 30, page 13) remains too long and unbalances the overall structure of the paper. I agree with Reviewer 2 on this point although reducing everything down to largely the page 9 bullet points is not reasonable. But could you aim for one page on NCE and one page on BPE please? The other major point raised by Reviewer 2 is the question of equivalency between energy and money and energy and inflation relations (particularly around pages 13-14). That material does need some attention. It would be helpful to move the coastal justifications from the ends of paragraphs to the beginnings. The marked PDF helpfully highlights some repetition, unfinished sentences, lapses in style and shifts in tense which need attention – but these changes should not take up too much time.

---

## [Reviewer Report]

For the most part, I accept the edits/changes by the authors...they have improved the manuscript considerably.

---

## [Editor Report]

I have been through the response to reviewer comments alongside the revised manuscript. The review of R1 highlighted the need to shorten the section on neoclassical economics and BPE. This has been done; the section is now 50% shorter than its previous length, without loss of impact. The review of R1 also highlighted the difficulty of the equivalence between energy and money. All this problematic text has been removed. These changes significantly rebalance the paper into a much more readable and better argued whole. In addition, some 16 more minor points have been attended to in full. I am completely satisfied that this is a serious and constructive revision. My decision is ‘accept’.